# Why did some parents not send their children back to school following school closures during the COVID-19 pandemic: a cross-sectional survey

Lisa Woodland  ,[1] Louise E Smith,[1] Rebecca K Webster,[2] Richard Amlôt,[3] Antonia Rubin,[4] Simon Wessely,[1] James G Rubin[1]

[1]Institution of Psychiatry Psychology and Neuroscience, King's College London, London, UK
[2]Psychology Deparment, The University of Sheffield, Sheffield, UK
[3]Public Health England, Salisbury, UK
[4]Weald of Kent Grammar School, Tonbridge, UK

**Correspondence to**
Lisa Woodland; lisa.woodland@kcl.ac.uk

## ABSTRACT

**Background** On 23 March 2020, schools closed to most children in England in response to COVID-19 until September 2020. Schools were kept open to children of key workers and vulnerable children on a voluntary basis. Starting 1 June 2020, children in reception (4–5 years old), year 1 (5–6 years old) and year 6 (10–11 years old) also became eligible to attend school.

**Methods** 1373 parents or guardians of children eligible to attend school completed a cross-sectional survey between 8 and 11 June 2020. We investigated factors associated with whether children attended school or not.

**Results** 46% (n=370/803) of children in year groups eligible to attend school and 13% (n=72/570) of children of key workers had attended school in the past 7 days. The most common reasons for sending children to school were that the child's education would benefit, the child wanted to go to school and the parent needed to work. A child was significantly more likely to attend if the parent believed the child had already had COVID-19, they had special educational needs or a person in the household had COVID-19 symptoms.

**Conclusions** Following any future school closure, helping parents to feel comfortable returning their child to school will require policy makers and school leaders to communicate about the adequacy of their policies to: (A) ensure that the risk to children in school is minimised; (B) ensure that the educational potential within schools is maximised; and (C) ensure that the benefits of school for the psychological well-being of children are prioritised.

### What is known about the subject?

► The COVID-19 pandemic resulted in the mass closure of schools for an extended period of time. A previous systematic review assessed parental attitudes towards smaller scale closures, finding that many parents (71%–97% across six studies) approved of closures, particularly where they were seen as an effective protective measure against a serious illness. The effect on a child's education and ability of the parent to work were reported as concerns.

### What this study adds?

► During the partial reopening of schools in England in June 2020, most parents did not send their children to school. Parents who were not educated to degree level, not working, who lived in the North of England or who were from black, Asian and ethnic minority backgrounds were least likely to send their children back. Perceived benefits of education, risk of disease and children's well-being were the main drivers in determining parental decision to send them to school or not.

## INTRODUCTION

On 23 March 2020, a nationwide closure of schools occurred across England in response to the COVID-19 pandemic. Only vulnerable children (those with a healthcare plan or social worker) and children of key workers (critical to the COVID-19 response) were able to attend.[1 2] From 1 June 2020, children in reception (4–5 years), year 1 (5–6 years) and year 6 (10–11 years) also became eligible to attend.[3] Until September 2020, school attendance was voluntary to those children eligible to attend.[4]

The benefits of closing schools to reduce the transmission of COVID-19 and the negative consequences of doing so were difficult to balance.[5 6] Adding to the debate was emerging evidence of a low transmission rate of COVID-19 among children[7–10] and a recognition that outbreaks may occur nonetheless.[11 12]

Irrespective of this debate, it was clear that many parents felt far from comfortable with their children attending school in the first months of the pandemic, even where it was encouraged.[13] A worldwide systematic review of school closures suggests several factors that may be relevant to whether a child attended school during an infectious disease outbreak.[14] Nineteen papers were included in

the review, samples representing between 67 and 4171 school-aged children (5–19 years). Perceived risk of infection,[15 16] concern about the impact of a closure on education[15 17] and parental concerns about their child's mental health were key issues.[18] Understanding the key issues that determine whether a parent is willing to send their children back to school when it is partially open, and ensuring that school policies and communications address these concerns, should help inform reopening schools, in this or any future pandemic.

In this study, we investigated factors associated with a parent's willingness to send their child to school when they partially reopened, following closures due to the COVID-19 pandemic. We investigated these factors for children in reception, year 1 or year 6 and for families where at least one parent was a key worker.

## METHODS

### Design

We commissioned a market research company, BMG Research, to administer a cross-sectional survey 1 week after schools in England reopened for children in reception, year 1 and year 6 (8–11 June 2020).[19] We have previously reported data from this survey relating to parental perceptions of the presence of hygiene procedures within schools.[20]

### Participants

Participants (n=2447) were recruited from BMG Research's panel, and to achieve a sample broadly representative of the population, BMG Research monitored region, child age, child gender, parent/guardian age and parent/guardian gender. Participants were eligible for the study if they were aged 18 years or over, lived in England and were a parent or guardian to a school-aged child (4–18 years) who usually lived with them. One hundred and eighty-three participants were screened out for non-eligibility, 226 participants dropped out after starting the survey and 28 completed but were removed for quality control such as completing the survey quickly or for 'straight-lining' (selecting the same option for every question) suggesting inattention to the questions. A total of 2010 participants remained. The sample fell within five percentage points of the national population by the child's gender, key stage and type of school attended against the known distribution for school children in England.[21] The sample used in this paper were 803 parents of children in eligible school years and 570 parents from families in which at least one parent was a key worker (nine participants were removed from this group due to logical inconsistencies which suggested they had accidently completed the wrong section).

Participants were paid equivalent to £0.60.

### Study materials

The full survey is available in the online supplemental materials.

All participants answered questions referring to their child who had the most recent birthday. In cases where children shared a birthday, we asked the parent to select one child.

Our survey had two sections. Section 1 was only completed by parents who had a child in reception, year 1 or year 6 or by parents who did not have a child in these year groups, but they or their spouse were a key worker. It contained questions about whether the child had attended school in the past week. Section 2 was completed by these parents and also by parents who did not have a child eligible to attend school. It contained general questions on views about risk of COVID-19, family living and school safety measures. In this paper, we only report data from section 1, relating to actual attendance in the past week.

### Personal characteristics

We asked participants to report their gender, age, region, household income, employment status, marital status, ethnicity and level of education. We also asked whether anyone within the household was aged over 70 years or had a health condition that made them vulnerable to COVID-19.

We asked participants to report the child's gender, age, school year, school type (fee paying or state funded) and whether the child had special educational needs (SENs).

### School attendance

Participants were asked how many times the child had attended school in the past 7 days. Depending on the response, parents were presented with randomised statements: 10 for why they were sending the child to school; 12 for why they were only sending the child to school part-time; or 16 why their child was not attending school. We asked participants to 'tick any [statement] that applies'. Participants also had the option to write-in text for 'other reason'.

### COVID-19 symptoms

We asked participants to report if the selected child had experienced any symptoms 'in the past 7 days' from a list of 10 symptoms. We also asked if they or a household member (other than the child) had experienced symptoms 'in the past 14 days' from the same symptom list. We asked participants whether they thought their child had had COVID-19.

### Well-being

We asked participants to report the child's well-being using two subscales from the Revised Child Anxiety Disorder Scale (RCADS):[22] the generalised anxiety disorder (GAD) subscale and the major depressive disorder (MDD) subscale.

### Patient and public involvement

A school trustee contributed to the development of the survey materials and coauthored this paper.

## ANALYSIS
### Recoding of variables
School attendance was defined as a child who attended school for at least 1 day in the past 7 days.

We created two binary variables to indicate whether the child, and someone in the household (other than the child), had a health condition that might make them particularly vulnerable to COVID-19.

We created two binary variables to indicate recent presence of COVID-19 symptoms in the child and someone else in the household. We defined presence of COVID-19 symptoms as experiencing a 'new, continuous cough', 'high temperature/fever', 'loss of sense of smell (fully or partial)' or 'loss of taste'. We coded a binary variable for the parent's perception of whether the child had had COVID-19 by grouping together 'they have definitely had it or definitely have it now' and 'they have probably had it or probably have it now'.

We created a binary variable to indicate low well-being in the child. We assigned a value against each answer from 0 ('never') to 3 ('always') on the GAD and MDD RCADS subscales and created a total score for each subscale. We then turned each total score into a t-score, a method used to normalise RCADS scores within the population, by child's age and gender.[22] [23] We used the same process for reception to year 3 as for year 4. We used a t-score cut-off of 65 or above on either GAD or MDD subscales to indicate low well-being.

For all variables, we coded the responses 'don't know', 'not applicable', 'prefer not to say' and 'prefer to self-describe' as missing data.

### Analysis
We ran a series of binary logistic regressions using SPSS V.26.0,[24] investigating univariable associations between each of our predictor variables and sending the child to school. We ran a second set of binary logistic regressions controlling for personal characteristics shown in the results.

We analysed frequencies for the statements responding to sending the child to school for a full week (5 days), part-time (less than 5 days) and for not sending the child to school.

For ease of interpretation, we used unweighted data in our analysis.

We applied a Bonferroni correction to our results (p≤0.001) due to running many analyses (children in school years n=29 and children of key workers n=31). Results meeting this criterion are marked by a dagger (†) in the tables.

### Sample size calculation
Post hoc calculations were run on the two subsamples that had a margin of error of plus or minus 5% at the 95% confidence level for each prevalence estimate.

## RESULTS
### School attendance
Nearly half (46%, 95% CI 43% to 50%, n=370/803) of children in eligible school years had attended school and about half had not (54%, 95% CI 50% to 57%, n=432/803). One participant was unsure if the child had attended school. The most common reasons for not sending a child to school were: thinking it was too risky for the child to attend school (n=223, 52%), the school not being open (n=140, 32%) and having another child who could not go to school (n=67, 16%). The most common reasons for sending a child to school were: the child's education would benefit (n=208, 56%), the child wants to go to school (n=200, 54%) and the child will benefit from seeing their friends (n=187, 51%). Of participants whose child did not attend school for a full week (n=204, 25%), the most common reasons for partial attendance were: the school only offers them to be in part-time (n=80, 39%), it is less risky for them to be in part-time (n=40, 20%) and parent only sending them in on days where the lessons are important (n=28, 14%).

Only 13% (95% CI 10% to 15%, n=72/570) of children of key workers had attended school, most children had not (87%, 95% CI 85% to 90%, n=497/570). One participant was unsure if the child had attended school. The most common reasons for children not attending school were: the school was not open (n=259, 52%), the school had asked the child not to attend (n=117, 24%) and thinking that it was too risky for the child to attend school at the moment (n=109, 22%). The most common reasons for attending school were: the parent needing to work (n=40, 56%), the child wanting to go to school (n=35, 49%) and thinking that the child's education would benefit from being at school (n=30, 42%). Of participants with children who attended school part-time, the most common reasons were: the school only offered them to be in part-time (n=16, 41%), parent only sent them in on days where the lessons are important (n=8, 21%) and believing it was less risky for them to be in part-time (n=7, 18%).

### Associations
Participant characteristics for children in eligible school years and for children of key workers are shown in table 1. Parents of children in eligible school years were less likely to send the child to school if they were educated to A-level or below, not working, of black, Asian and minority ethnicity (BAME) or living in the North East, North West and Yorkshire and the Humber compared with London, whereas children of key workers were more likely to attend school for participants aged 45 years and under and who were working. Child attendance was more likely for children in eligible school years when in a fee-paying school and if they had a health condition that made them vulnerable to COVID-19. For both groups, attendance was more likely for children who had SENs, reported low well-being, thought their child had had COVID-19 and when a person over 70 years was living in the household. There was also a significant association in both groups between a child being more likely to attend school and the child having experienced COVID-19 symptoms in the past 7 days or another person in the household having experienced COVID-19 symptoms in the past 14 days.

**Table 1** Participant characteristics for children in eligible school years (n=803) and children of key workers (n=570) by school attendance

| | Level | Children in eligible school years | | | | Children of key workers | | | |
|---|---|---|---|---|---|---|---|---|---|
| | | Did not attend school, n (%) | Attended school, n (%) | OR (95% CI) | Adjusted OR (95% CI)‡ | Did not attend school, n (%) | Attended school, n (%) | OR (95% CI) | Adjusted OR (95% CI)‡ |
| Gender | Male | 201 (52) | 187 (48) | 1.22 (0.92 to 1.61) | 1.04 (0.75 to 1.43) | 227 (85) | 39 (15) | 1.53 (0.92 to 2.55) | 1.61 (0.91 to 2.85) |
| | Female | 230 (57) | 177 (43) | Reference | Reference | 268 (90) | 30 (10) | Reference | Reference |
| Age (years) | 18–35 | 129 (49) | 136 (51) | 1.37 (0.92 to 2.03) | 1.23 (0.79 to 1.92) | 54 (77) | 16 (23) | 3.16* (1.56 to 6.37) | 2.95* (1.23 to 7.08) |
| | 36–45 | 212 (56) | 164 (44) | 1.01 (0.70 to 1.46) | 0.98 (0.65 to 1.48) | 198 (86) | 33 (14) | 1.76* (1.01 to 3.12) | 2.14* (1.12 to 4.09) |
| | More than and equal to 46 years | 91 (56.5) | 70 (43.5) | Reference | Reference | 245 (91) | 23 (9) | Reference | Reference |
| Region | East Midlands | 33 (60) | 22 (40) | 0.50* (0.27 to 0.93) | 0.59 (0.31 to 1.14) | 45 (85) | 8 (15) | 0.85 (0.32 to 2.22) | 1.23 (0.40 to 3.75) |
| | East of England | 38 (48) | 41 (52) | 0.81 (0.48 to 1.38) | 0.92 (0.52 to 1.64) | 54 (82) | 12 (18) | 1.06 (0.45 to 2.52) | 1.38 (0.50 to 3.81) |
| | North East | 33 (67) | 16 (33) | 0.36* (0.19 to 0.71) | 0.38* (0.18 to 0.78) | 32 (89) | 4 (11) | 0.60 (0.18 to 1.98) | 0.93 (0.25 to 3.46) |
| | North West | 68 (65) | 36 (35) | 0.40† (0.24 to 0.66) | 0.39* (0.23 to 0.68) | 80 (93) | 6 (7) | 0.36* (0.13 to 0.99) | 0.61 (0.19 to 1.89) |
| | South East | 70 (51) | 66 (49) | 0.71 (0.45 to 1.11) | 0.92 (0.56 to 1.52) | 82 (89) | 10 (11) | 0.58 (0.24 to 1.94) | 0.81 (0.29 to 2.29) |
| | South West | 24 (46) | 28 (54) | 0.88 (0.47 to 1.63) | 1.16 (0.58 to 2.31) | 42 (87.5) | 6 (12.5) | 0.68 (0.42 to 1.94) | 1.18 (0.37 to 3.82) |
| | West Midlands | 39 (56) | 30 (44) | 0.58 (0.33 to 1.01) | 0.63 (0.34 to 1.18) | 48 (89) | 6 (11) | 0.60 (0.21 to 1.68) | 0.84 (0.27 to 2.60) |
| | Yorkshire and the Humber | 52 (62) | 31 (37) | 0.45* (0.26 to 0.76) | 0.52* (0.29 to 0.93) | 52 (88) | 7 (12) | 0.64 (0.24 to 1.73) | 0.80 (0.27 to 2.41) |
| | London | 75 (43) | 100 (57) | Reference | Reference | 62 (83) | 13 (17) | Reference | Reference |
| Household income | ≤£34 999 | 164 (54 | 142 (46 | 1.00 (0.75 to 1.33) | 1.34 (0.95 to 1.90) | 185 (89) | 23 (11) | 0.75 (0.44 to 1.28) | 0.96 (0.51 to 1.81) |
| | ≥£35 000 | 254 (54) | 220 (46) | Reference | Reference | 285 (86) | 47 (14) | Reference | Reference |
| Employment status§ | Working | 362 (52) | 337 (48) | 1.95* (1.25 to 3.05) | 1.94* (1.16 to 3.24) | 435 (86) | 70 (14) | 9.33* (1.27 to 68.47) | 8.64* (1.13 to 65.96) |
| | Not working | 67 (68) | 32 (32) | Reference | Reference | 58 (98) | 1 (2) | Reference | Reference |
| Education level | ≤A-level | 179 (63) | 105 (37) | 0.56† (0.41 to 0.75) | 0.59* (0.42 to 0.84) | 227 (88) | 31 (12) | 0.91 (0.55 to 1.51) | 1.27 (0.71 to 2.26) |
| | ≥Degree | 248 (49) | 261 (51) | Reference | Reference | 267 (87) | 40 (13) | Reference | Reference |
| Marital status | Living alone | 49 (48) | 54 (52) | 1.34 (0.88 to 2.02) | 1.30 (0.82 to 2.06) | 75 (88) | 10 (12) | 0.91 (0.45 to 1.85) | 0.92 (0.41 to 2.03) |
| | Married/cohabiting | 383 (55) | 316 (45) | Reference | Reference | 422 (87) | 62 (13) | Reference | Reference |
| Ethnicity | White | 355 (53) | 316 (47) | 1.23 (0.84 to 1.12) | 1.66* (1.07 to 2.58) | 450 (87) | 65 (13) | 1.04 (0.42 to 2.53) | 1.47 (0.53 to 4.13) |
| | BAME | 72 (58) | 52 (42) | Reference | Reference | 43 (78) | 12 (22) | Reference | Reference |
| Child gender | Boy | 230 (51.5) | 217 (48.5) | 1.25 (0.94 to 1.65) | 1.15 (0.85 to 1.56) | 244 (87) | 36 (13) | 1.04 (0.63 to 1.70) | 0.96 (0.56 to 1.66) |
| | Girl | 202 (57) | 153 (43) | Reference | Reference | 253 (87.5) | 36 (12.5) | Reference | Reference |

Continued

**Table 1** Continued

| | | Children in eligible school years | | | | Children of key workers | | | |
|---|---|---|---|---|---|---|---|---|---|
| | Level | Did not attend school, n (%) | Attended school, n (%) | OR (95% CI) | Adjusted OR (95% CI)‡ | Did not attend school, n (%) | Attended school, n (%) | OR (95% CI) | Adjusted OR (95% CI)‡ |
| School year | Early years | 77 (59) | 53 (41) | 0.74 (0.50 to 1.10) | 0.79 (0.51 to 1.23) | ¶ | ¶ | ¶ | ¶ |
| | Key stage 1 | 157 (54) | 132 (46) | 0.90 (0.66 to 1.22) | 0.88 (0.63 to 1.24) | 30 (86) | 5 (14) | 2.10 (0.57 to 7.81) | 1.52 (0.37 to 6.30) |
| | Key stage 2 | 198 (52) | 185 (48) | Reference | Reference | 120 (86) | 20 (15) | 2.10 (0.75 to 5.86) | 1.45 (0.47 to 4.44) |
| | Key stage 3 | ¶ | ¶ | ¶ | ¶ | 152 (87) | 23 (13) | 1.91 (0.69 to 5.24) | 1.58 (0.55 to 4.55) |
| | Key stage 4 | ¶ | ¶ | ¶ | ¶ | 132 (87) | 19 (13) | 1.81 (0.65 to 5.08) | 1.44 (0.49 to 4.24) |
| | Years 12 and 13 | ¶ | ¶ | ¶ | ¶ | 63 (93) | 5 (7) | Reference | Reference |
| School type | Fee paying | 35 (33) | 71 (67) | **2.68† (1.74 to 4.13)** | **2.50* (1.56 to 4.00)** | 30 (77) | 9 (23) | **2.20* (1.00 to 4.85)** | 2.04 (0.82 to 5.07) |
| | State funded | 395 (57) | 299 (43) | Reference | Reference | 462 (88) | 63 (12) | Reference | Reference |
| Child has SEN | Yes | 22 (31) | 48 (69) | **2.78† (1.64 to 4.70)** | **2.30* (1.27 to 4.17)** | 28 (61) | 18 (39) | **5.74† (2.97 to 11.08)** | **7.92† (3.59 to 17.46)** |
| | No | 406 (56) | 319 (44) | Reference | Reference | 464 (90) | 52 (10) | Reference | Reference |
| Low child well-being | Yes | 92 (42) | 124 (57) | **1.86† (1.36 to 2.55)** | **1.47* (1.04 to 2.07)** | 127 (82) | 28 (18) | **1.85* (1.11 to 3.10)** | **1.78* (1.00 to 3.21)** |
| | No | 340 (58) | 246 (42) | Reference | Reference | 370 (89) | 44 (11) | Reference | Reference |
| Child vulnerable COVID-19 | Yes | 28 (39) | 44 (61) | **2.00* (1.21 to 3.28)** | **1.76* (1.00 to 3.10)** | 33 (82.5) | 7 (17.5) | 1.49 (0.63 to 3.50) | 0.93 (0.33 to 2.63) |
| | No | 401 (56) | 316 (44) | Reference | Reference | 455 (87.5) | 65 (12.5) | Reference | Reference |
| Household vulnerable COVID-19 | Yes | 115 (60) | 78 (40) | 0.78 (0.56 to 1.08) | 0.74 (0.51 to 1.06) | 134 (88) | 18 (12) | 0.90 (0.50 to 1.60) | 0.56 (0.44 to 1.67) |
| | No | 295 (53) | 258 (47) | Reference | Reference | 314 (87) | 47 (13) | Reference | Reference |
| Child COVID-19 symptoms | Yes | 15 (24) | 47 (76) | **5.05† (2.22 to 7.36)** | **3.02* (1.49 to 6.15)** | 11 (58) | 8 (42) | **5.52† (2.14 to 14.24)** | **5.25* (1.58 to 17.41)** |
| | No | 417 (56) | 323 (44) | Reference | Reference | 486 (88) | 64 (12) | Reference | Reference |
| Household COVID-19 symptoms | Yes | 9 (17) | 43 (83) | **6.18† (2.97 to 12.86)** | **6.04† (2.62 to 13.91)** | 13 (65) | 7 (35) | **4.00* (1.54 to 10.42)** | **4.11* (1.26 to 13.44)** |
| | No | 423 (56) | 327 (43) | Reference | Reference | 484 (88) | 65 (12) | Reference | Reference |
| Child has had or currently has COVID-19 | Yes | 36 (24) | 112 (76) | **4.78† (3.18 to 7.17)** | **4.03† (2.57 to 6.30)** | 40 (67) | 20 (33) | **4.40† (2.40 to 8.08)** | **5.20† (2.48 to 10.93)** |
| | No | 396 (61) | 258 (37) | Reference | Reference | 457 (90) | 52 (10) | Reference | Reference |
| Someone over 70 years | Yes | 11 (26) | 32 (74) | **3.62† (1.80 to 7.30)** | **3.19* (1.45 to 7.05)** | 8 (57) | 6 (43) | **5.56* (1.87 to 16.52)** | 3.49* (0.98 to 12.46) |
| | No | 421 (55.5) | 338 (44.5) | Reference | Reference | 489 (88) | 66 (12) | Reference | Reference |

Decimal places have been used for rounding errors.
*P≤ 0.05 and formatted bold.
†P≤ 0.001 and formatted bold.
‡Adjusting for participant gender, age, region, household income, employment status, education level, marital status, ethnicity and the child's gender, school year and school type.
§Working includes students and volunteers.
¶Data not applicable for the sample.
BAME, black, Asian and minority ethnicity; SEN, special educational need.

## DISCUSSION

Most children eligible to attend school did not attend 1 week after schools in England began to reopen to more children.[25] Worryingly, we observed patterns that seemed likely to entrench existing educational inequalities. Children from households where parents have lower education achievements, BAME households and households in the North of England and state funded being least likely to attend school.

Our results suggest that several broad areas determined attendance at school. First, risk perceptions were crucial. The response 'it is too risky' featured highly in parental reasons for not sending children to school. Second, children were also more likely to attend school when parents thought their child had COVID-19 which could indicate the belief their child is immune to further infection.[26] Unexpectedly, children were more likely to attend school when they had a health condition that made them vulnerable to COVID-19 and when a person over 70 years was living in the household. We speculate that this may be because these circumstances affect the parent's ability to look after the child at home.

Poorer perceived child well-being was associated with them being more likely to attend school. This finding may reflect parental desire to improve their child's well-being that may have been impacted by the school closure.[27] This was also apparent in reasons parents gave for sending their child to school. As expected,[14] concerns about education also featured highly as a reason for attendance, while perceptions that schools could not provide good quality education or that some lessons were not as important were cited as reasons for absence or partial attendance.

Despite most children of key workers not attending school, needing to work was the most reported reason for school attendance, and the school not opening or asking the child not to attend were the most commonly reported reasons for the child not being in school. This suggests that work commitments were the main driver for this group.

One notable finding was that children who had symptoms of COVID-19 in the past 7 days or whose household members had these symptoms in the past 14 days were significantly more likely to have gone to school. We do not know if the child attended school while having symptoms or when symptoms were present in the household (against self-isolation guidance).[28] However, given that school closure is specifically designed to reduce the transmission of respiratory infections in general, it is also possible that this reflects the re-emergence of transmission of upper respiratory tract infections, or COVID-19 specifically, within the school environment.[29] It may also reflect increased anxiety or awareness among parents around COVID-19 symptoms, resulting in higher symptom detection and reporting.

### Limitations

Several limitations should be borne in mind for this study. First the cross-sectional nature of this study limits our ability to draw causal findings. Second, the RCADS subscale was used to indicate low well-being but is currently not validated for children under 8 years.[30] Third, online polls can be unrepresentative and lead to response and self-reporting bias.[31] However, in line with the reasoning relating to the use of non-probability samples in social sciences,[32] we assume that the associations within our data do generalise to the wider population. Fourth, we ran many analyses raising the possibility of type 1 errors. While we have provided Bonferroni corrections in the tables for readers who wish to correct this, this correction in turn may be overly conservative.

## CONCLUSION

Our findings support previous research by suggesting that during an infectious disease outbreak, parents' decision to send their child to school was impacted by the risk of disease (COVID-19), child's education and well-being. Furthermore, without reassuring parents in these three areas and encouraging them to send their children to school health inequalities are likely to be increased.

**Contributors** All authors contributed to the conceptualisation of the study and approved the final draft. LW designed the survey, analysed the data and drafted the manuscript. LES and GJR designed the survey, analysed the data and edited the manuscript. RKW edited the manuscript. RA and AR designed the survey and edited the manuscript. SW edited the manuscript.

**Funding** This study was funded by the Economic and Social Research Council [grant number ES/P000703/1] and the National Institute for Health Research Health Protection Research Unit (NIHR HPRU) [grant number NIHR200890] in Emergency Preparedness and Response, a partnership between Public Health England, King's College London and the University of East Anglia.

**Disclaimer** The views expressed are those of the authors and not necessarily those of the NIHR, Public Health England or the Department of Health and Social Care. The funders had no role in study design, data collection, data analysis, data interpretation or writing of the manuscript. The corresponding author had full access to all the data and had final responsibility for the decision to submit for publication.

**Competing interests** None declared.

**Patient consent for publication** Not required.

**Ethics approval** The research was approved by the Psychiatry, Nursing and Midwifery Research Ethics Subcommittee at King's College London (LRS—19/20–18787).

**Provenance and peer review** Not commissioned; externally peer reviewed.

**Data availability statement** The full survey and reasons for child's school attendance (frequency tables) are available in the supplemental materials. We are unable to share all data due to data confidentially. Further materials may be available upon request to the corresponding author.

**ORCID iD**
Lisa Woodland http://orcid.org/0000-0003-2440-3210

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
