## [Reviewer comments · BMJ Paediatrics Open]

ARTICLE DETAILS

TITLE (PROVISIONAL)	Why did some parents not send their children back to school following school closures during the COVID-19 pandemic: a cross-sectional survey
AUTHORS	Woodland, Lisa Smith, Louise E Webster, Rebecca K Amlôt, Richard Rubin, Antonia Wessely, Simon Rubin, James G

VERSION 1 – REVIEW

REVIEWER	Reviewer name: Dr. Sunil Bhopal Institution and Country: Newcastle University, United Kingdom of Great Britain and Northern Ireland Competing interests: None
REVIEW RETURNED	26-Jan-2021

GENERAL COMMENTS	Dear Authors, Thank you for giving me the opportunity to review this interesting paper. It poses the question "what factors were associated with parents being willing to send their children back to school in England in early June 2020" when Covid-19 case numbers, hospitalisations and deaths had fallen dramatically from March/April. You surveyed parents in early June 2020 to help understand parental willingness and concerns about returning their children to in-school education. Here are some suggestions for improvement which I hope you find useful: INTRODUCTION Ref 1: suggest reference government document rather than speech as this is more detailed, and define this in text including definition of 'keyworker' Paragraph 1: It's relatively clear to me because I was involved, but think this needs some work to bring out clarity for someone who wasn't there. Perhaps, "from x March to y June ONLY children of..." were eligible. Line 20: define early stages, or state dates? Lines 21-28 need fleshing out: what type of studies are these references, how many individuals, what did they say, what were the findings, what was the context (age of child, geography etc), I find it too brief and non informative at the moment Lines 34-36: To make this as useful for an international audience as
---

possible suggest naming ages again

METHODS

Line 44: can you explain 'commissioned' here, i.e you didn't do the survey?

Line 51: who is BMG?

Line 58: Suggest referencing rationale for rejecting 'straightlining' for those unfamiliar with this

Line 59: "This left" could be rephrased

Line 59-60 "Monitored by the panel to..." - think need to spell out the aim first, i.e to get a sample representative of UK population with respect to x,y,z and then explain the method of doing so

Page 6, line 9: I am not sure about this wording "earned points from the research panel" - not clear to me what a 'research panel' is

Page 6, line 14: Could you state what participants did first and then if still need to mention pseudorandomly say that afterwards - or just delete? Consider what it adds?

Page 6, lines 18-24: Could you reword for clarity

Page 6, line 27: Is this a sub-section? Or do the subheadings e.g personal characteristics fall under the subheading Study Materials? This seems unnecessarily short at present

Ethics p6line20 - could add informed consent details here

NUMBERS OF PARTICIPANTS: this is mentioned in methods p4line51-60 and again in Sample Size Calculation p6line55-59: suggest all of this is consolidated. In methods I suggest you state the intention and in results the actual numbers with reasons for exclusions etc

Sample size calculation: is this a post-hoc power calculation based on numbers recruited, and the results obtained? I suspect not, but it reads like that. I wonder if you might more simply state that a sample of X gives 90% power to calculate a 95%CI for prevalence estimates of X% or similar? You may want to tell us how this was done. See what you think.

p7lines 11-13 suggest splitting this sentence up to aid clarity

p7line 14 - does this 'only' help here? could this be stated without?

p7 lines 11-24: suggest this paragraph needs some work for clarity e.g "the most common reason" leads on from an unrelated sentence

TABLES:

"Children in schools years" suggest this column header needs renaming

In methods it mentions a likert-type scale from strongly dis, to strongly agree - but in the tables % are presented - can you clarify Table 1- Should 'other' be at the bottom of table

Is it necessary to list the covid symptoms in the table rows? Would neaten up to remove

Table headers - suggest this needs rewording to make it stand alone. At present "participant reasons for children eligible for school" doesn't have meaning to me

Tables 1-3: These feel repetitive and it's difficult to read across them to compare answers between groups. I would suggest giving

	some thought to how you could combine these tables. Perhaps (depending on the answer to the question re Likert scale above) you could do colour coding, add more columns to the table, or come up with another solution that helps the reader to see the differences you found Table 6: Can you explain in what ways "Want to send to school" and "If it were possible, I would feel comfortable sending child to school next week". I see the very strong association and large OR. DISCUSSION Sentence one needs to be fleshed out to give more context, timings, country, etc Sentence two contains a double-negative Paragraph 2 contains lots of results and needs more interpretation to make it truly as part of the 'discussion' as you have done in paragraph 3. LIMITATIONS I would suggest starting with the simpler point that this is a cross-sectional study, using new (unvalidated) questions, then move on to the sampling methods, causality, and statistics at the end CONCLUSION Sentence 1 does not stand alone. The final sentence feels like an overreach from the data presented. I would suggest that this section needs some fleshing out and reworking QUESTIONNAIRE It would be worth mentioning in the main text that the statements were randomised Recoding of variables section This should be in methods Small further comments: How many twins were there (same dates of birth)? Suggest check throughout for numbers written in full that could be in numerals e.g p5 line 57 Very best wishes.
--	---

REVIEWER	Reviewer name: Dr. Peter Flom Institution and Country: Peter Flom Consulting, United States Competing interests: None
REVIEW RETURNED	06-Jan-2021

GENERAL COMMENTS	I confine my remarks to statistical aspects of this paper. Unfortunately I think there are some fairly major issues. The way the questionnaires were scored is problematic. The Likert scale items should not be dichotomized (indeed, no multi-category answer should be). Ideally, the items would be factor analyzed (using methods appropriate for ordinal variables) and the resultant scores used as continuous variables. Other issues: How were the scales validated? Were they pilot tested? Were they tested for reliability?
---

	p. 5 How ere the variables operationalized? p. 6 The analysis should include ordinal logistic (for ordinal variables) and OLS regression (for the ones that use factor scores, from above).
--	---

REVIEWER	Reviewer name: Dr. Tim Cheetham Institution and Country: Newcastle Upon Tyne Hospitals NHS Foundation Trust, United Kingdom of Great Britain and Northern Ireland Competing interests: None
REVIEW RETURNED	08-Feb-2021

GENERAL COMMENTS	The authors wanted to establish why parents sent or did not send their child back to school during the COVID pandemic.  1. Was there anything in the conclusions section of the abstract that you found surprising? What you have stated here seems to be sound principles irrespective of COVID –what is actually new here? 2. Likewise the final sentence in the conclusion ‘Parents will feel more comfortable in sending their children to school if reassured that school will be safe, educational and enjoyable’ – is this stating the obvious? 3. When you started the study – did you have a hypothesis? 4. Design – why 1 week after schools reopened? – why not 2 weeks? – or 2 days? 5. Participants – what do the initials BMG stand for? 6. Sample size calculation – it sounds like this was conducted after you had collected data rather than beforehand – is this correct? 7. What is meant by ‘participants earned points from the research panel?’ – more detail please so it is possible to understand what actually happened. 8. How were the statements devised? 9. PPI – do you think that one school trustee was in a position to provide pertinent feedback? Who was the trustee? Were they a parent of school-age children? Why not tap into the thoughts of a larger group of parents? I don’t understand why the questions were based on the opinion of one person alone... 10. How long does it take to fill in the survey? – what did you tell potential participants about this? (see supplementary material). The process looks pretty onerous to me...has this got any implications for how the form was completed? 11. The key information is contained in the lengthy, detailed tables and I wondered if the authors can think of an approach that might make it easier to digest this information. Would it be helpful to summarise the ‘positive’ findings in a separate table, for example? 12. Table 1 – what about ‘ I was waiting to see how other children got on first’ – is this a pertinent reason? 13. Why did the children of keyworkers not attend school in the main?
---

VERSION 1 – AUTHOR RESPONSE

Wednesday, 29 September 2021

Dear Prof. Choonara,

We have updated our manuscript from recommendations of your team and peer reviewers below and highlighted yellow in the manuscript. We would like to thank all

reviewers for their time, and we hope that we have adequately responded to all comments.

Reviewer(s)' Comments to Author (if any):

Reviewer: 1

Reviewer name: Peter Flom

Reviewer: 2

Reviewer name: Sunil Bhopal

Reviewer: 3

Reviewer name: tim cheetham

Reviewer: 1

Institution and Country: Peter Flom Consulting, USA

Reviewer: 2

Institution and Country: Newcastle University, UK

Reviewer: 3

Institution and Country: Newcastle University

Reviewer: 1

Comments to the Author

I confine my remarks to statistical aspects of this paper. Unfortunately I think there are some fairly major issues.

1. The way the questionnaires were scored is problematic. The Likert scale items should not be dichotomized (indeed, no multi-category answer should be). Ideally, the items would be factor analyzed (using methods appropriate for ordinal variables) and the resultant scores used as continuous variables.

Response: Thank you for the comments. After discussing your comment with a statistician we feel that the data regarding intention of what parents might do next week (which used the Likert scale and would require factor analysis) is of lesser interest to readers than our data relating to what parents actually did do last week. We have therefore removed the section relating to future intentions and focus instead on parents actual behaviour. This has simplified our message, while retaining all of the key conclusions. In liaison with our statistician, we hope to present the data on future intentions elsewhere, and for now have simply explicitly flagged to the readers that these data were collected.

Other issues:

2. How were the scales validated? Were they pilot tested? Were they tested for reliability?

Response: Having removed the analyses about future intentions, many of these issues no longer apply. In terms of our main outcome measure "have they attended school at all in the past seven days," we feel this has sufficient face validity. All questions are now presented in the supplementary materials to allow readers to review them in detail. No scales were used, other than those relating to child wellbeing – the references for the validation of these are given. All other variables used consisted of single items with good face validity.

3. How were the variables operationalized?

Response: Thank you for the comment. The re-coding of variables has now been described in the main text (Methods section, Recoding of variables).

4. The analysis should include ordinal logistic (for ordinal variables) and OLS regression (for the ones that use factor scores, from above).

Response: The removal of analyses relating to future intentions should have resolved this issue – no ordinal outcome measures are now reported.

Reviewer: 2

Comments to the Author

Dear Authors,

Thank you for giving me the opportunity to review this interesting paper. It poses the question "what factors were associated with parents being willing to send their children back to school in England in early June 2020" when Covid-19 case numbers, hospitalisations and deaths had fallen dramatically from March/April.

You surveyed parents in early June 2020 to help understand parental willingness and concerns about returning their children to in-school education.

Here are some suggestions for improvement which I hope you find useful:

INTRODUCTION

5. Ref 1: suggest reference government document rather than speech as this is more detailed, and define this in text including definition of 'keyworker'

Response: Thank you for the comments. The government documents (web pages) are regularly updated due to changes in guidance. Therefore, they do not reflect the rules and regulations at the time of data collection. We used the government media announcements because they are static references. However, we have used detailed / official documents where appropriate. We have added the definition of key worker and included a government document for an additional reference.

6. Paragraph 1: It's relatively clear to me because I was involved, but think this needs some work to bring out clarity for someone who wasn't there. Perhaps, "from x March to y June ONLY children of..." were eligible.

Response: We have now clarified this section and provided the relevant dates.

7. Line 20: define early stages, or state dates?

Response: We have changed this to "it is clear that many parents felt far from comfortable with their children attending school in the first months of the pandemic, even where it was encouraged"

8. Lines 21-28 need fleshing out: what type of studies are these references, how many individuals, what did they say, what were the findings, what was the context (age of child, geography etc), I find it too brief and non informative at the moment

Response: We have added more contextual information. "worldwide" and "19 papers were included in the review, samples representing between 67 and 4,171 school-aged children (five to 19 years)".

9. Lines 34-36: To make this as useful for an international audience as possible suggest naming ages again

Responses: Ages for these year groups appear where they are first discussed (paragraph one of the introduction). Given the word limit, we are unable to repeat information, but hope that, given the brevity of the introduction, readers will be able to find these ages relatively easily.

METHODS

10. Line 44: can you explain 'commissioned' here, i.e you didn't do the survey?

Response: You are correct in that we did not perform data collection. We designed the study materials but paid a market research company (BMG Research) to conduct the survey. We have now clarified this.

11. Line 51: who is BMG?

Response: We have now clarified that 'BMG Research' is a market research company.

12. Line 58: Suggest referencing rationale for rejecting 'straightlining' for those unfamiliar with this

Response: We have added to the definition of straightlining that it "suggests inattention to the questions."

13. Line 59: "This left" could be rephrased

Response: We have rephrased to "2,010 participants remained".

14. Line 59-60 "Monitored by the panel to..." - think need to spell out the aim first, i.e to get a sample representative of UK population with respect to x,y,z and then explain the method of doing so

Response: We have rephrased this sentence, which we hope is clearer.

15. Page 6, line 9: I am not sure about this wording "earned points from the research panel" - not clear to me what a 'research panel' is

Response: Research panel is the term used to describe the method of selecting participants (usually by demographic characteristics or knowledge of a particular topic / issue) from a pool of people who have been pre-recruited by a market research company to receive invitations about future studies. These participants are rewarded for taking part with points, which can be accumulated and exchanged for money once a certain threshold (e.g. £50) is reached. Rather than complicate the paper, and bearing the word limitations in mind, we have clarified that "participants were paid equivalent to £0.60."

16. Page 6, line 14: Could you state what participants did first and then if still need to mention pseudorandomly say that afterwards - or just delete? Consider what it adds?

Response: We have removed pseudorandomly.

17. Page 6, lines 18-24: Could you reword for clarity

Response: We have reworded this section which I hope brings clarity. "Our survey had two sections. Section one was only completed by parents who had a child in reception, year one, or year six or by parents who did not have a child in these year groups, but they or their spouse were a key worker. It contained questions about whether the child had attended school in the past week. Section two was completed by these parents and also by parents who did not have a child eligible to attend school. It contained general questions on views about risk of COVID-19, family living and school safety measures. In this paper we only report data from section one, relating to actual attendance in the past week."

18. Page 6, line 27: Is this a sub-section? Or do the subheadings e.g personal characteristics fall under the subheading Study Materials? This seems unnecessarily short at present

Response: We have made this section clearer to indicate each section are a sub-heading of study materials.

19. Ethics p6line20 - could add informed consent details here

Response: Unfortunately, given the word count restrictions we feel the confirming that the study met the ethical criteria for our institution is hopefully sufficient.

20. NUMBERS OF PARTICIPANTS: this is mentioned in methods p4line51-60 and again in Sample Size Calculation p6line55-59: suggest all of this is consolidated. In methods I suggest you state the intention and in results the actual numbers with reasons for exclusions etc

Response: We have consolidated this section.

21. Sample size calculation: is this a post-hoc power calculation based on numbers recruited, and the results obtained? I suspect not, but it reads like that. I wonder if you might more simply state that a sample of X gives 90% power to calculate a 95%CI for prevalence estimates of X% or similar? You may want to tell us how this was done. See what you think.

Response: We have altered the sample size section following comment 20 and therefore this section is now clearer. A post-hoc power calculation was used because we separated the data into sub-samples during analysis.

22. p7lines 11-13 suggest splitting this sentence up to aid clarity

Response: The results section has been updated and is hopefully now clearer.

23. p7line 14 - does this 'only' help here? could this be stated without?

Response: The results section has been updated and is hopefully clearer.

24. p7 lines 11-24: suggest this paragraph needs some work for clarity e.g "the most common reason" leads on from an unrelated sentence

Response: The results section has been updated and is hopefully clearer.

TABLES:

25. "Children in schools years" suggest this column header needs renaming In methods it mentions a likert-type scale from strongly dis, to strongly agree - but in the tables % are presented - can you clarify

Response: We used a Likert scale to measure willingness to send your child to school in the future. Following the statistical review comments, and discussion with our own statistician, we have now removed those data from this paper. We think this now tells a much cleaner story and hope it is now clearer that the outcome data we presented are dichotomous – the child either did, or did not, attend school last week.

26. Table 1- Should 'other' be at the bottom of table

Response: This was an error. Statement now states "other reason", these tables have also been moved to supplementary materials and used a more narrative approach as suggested in comment 30.

27. Is it necessary to list the covid symptoms in the table rows? Would neatening up to remove

Response: We have kept this in as it reflects the exact wording used and hopefully makes clear to the reader that we guided respondents as to what symptoms to consider.

28. Table headers - suggest this needs rewording to make it stand alone. At present "participant reasons for children eligible for school" doesn't have meaning to me

Response: We have clarified the table headings.

29. Tables 1-3: These feel repetitive and it's difficult to read across them to compare answers between groups. I would suggest giving some thought to how you could combine these tables. Perhaps (depending on the answer to the question re Likert scale above) you could do colour coding, add more columns to the table, or come up with another solution that helps the reader to see the differences you found

Response: We have now moved these tables into the supplementary materials and described the most interesting data narratively in the text.

30. Table 6: Can you explain in what ways "Want to send to school" and "If it were possible, I would feel comfortable sending child to school next week". I see the very strong association and large OR.

Response: As noted above, we have removed this section from the paper.

DISCUSSION

31. Sentence one needs to be fleshed out to give more context, timings, country, etc Sentence two contains a double-negative Paragraph 2 contains lots of results and needs more interpretation to make it truly as part of the 'discussion' as you have done in paragraph 3.

Response: We have removed the double negative sentence, reworded the opening sentence, and reviewed the discussion.

LIMITATIONS

32. I would suggest starting with the simpler point that this is a cross-sectional study, using new (unvalidated) questions, then move on to the sampling methods, causality, and statistics at the end

Response: We have re-ordered the limitations as suggested.

CONCLUSION

33. Sentence 1 does not stand alone. The final sentence feels like an overreach from the data presented. I would suggest that this section needs some fleshing out and reworking

Response: We have re-written the conclusion.

QUESTIONNAIRE

34. It would be worth mentioning in the main text that the statements were randomised

Response: Many of the relevant statements are no longer reported. However, we have added where relevant that some statements were presented in a random order.

35. Recoding of variables section, this should be in methods

Response: Thank you for the comment, we agree and have moved into the main paper.

Small further comments:

36. How many twins were there (same dates of birth)?

Response: We do not hold this information. We only asked participants to provide data about one child with the most recent birthday (and to select a child if two or more shared that birthday).

37. Suggest check throughout for numbers written in full that could be in numerals e.g p5 line 57

Response: Thank you for comment and we have resolved this.

Very best wishes.

Reviewer: 3

Comments to the Author

The authors wanted to establish why parents sent or did not send their child back to school during the COVID pandemic.

38. Was there anything in the conclusions section of the abstract that you found surprising? What you have stated here seems to be sound principles irrespective of COVID –what is actually new here?

Response: Thank you for your comments. We have re-worded our conclusion which we hope provides information on the results we found surprising and what was new.

39. Likewise the final sentence in the conclusion ‘Parents will feel more comfortable in sending their children to school if reassured that school will be safe, educational and enjoyable’ – is this stating the obvious?

Response: Having discussed these findings with policy makers, we can confirm that sometimes what can seem obvious to us, has a lot more impact with data behind it. We hope that a short paper emphasising that these points are empirically tested and confirmed will provide support to others having similar conversations now, or in future pandemics.

40. When you started the study – did you have a hypothesis?

Response: Our hypothesis is based on the systematic review we described in the introduction. Parental willingness to send their children to school would be impacted by perceived risk of infection, education and child’s mental health.

33. Design – why 1 week after schools reopened? – why not 2 weeks? – or 2 days?

Response: We could readily have chosen any period during this stage of the reopenings. However, we had to wait one week for our outcome measure to make sense (“have they attended school at all in the past seven days”).

34. Participants – what do the initials BMG stand for?

Response: We have responded to this in point 11 by reviewer 2. We have clarified the title of the company as ‘BMG Research’ and included a reference to their company website.

35. Sample size calculation – it sounds like this was conducted after you had collected data rather than beforehand – is this correct?

Response: Our sample size of about 2000 was chosen a priori to give a confidence interval of plus or minus about 2% for each prevalence estimate for the full sample. However in this paper we use smaller subsamples. We therefore use a post-hoc calculation on the sub samples. We also clarified this section in response to points 20 and 21 from reviewer 2.

36. What is meant by ‘participants earned points from the research panel? – more detail please so it is possible to understand what actually happened.

Response: Thank you for your comment. We clarified this following point 15 to reviewer 2. Research panel is the term used to describe the method of selecting participants (usually by demographic characteristics or knowledge of a particular topic / issue) from a pool of people who have been pre-recruited by a market

research company to receive invitations about future studies. These participants are rewarded for taking part with points, which can be accumulated and exchanged for money once a certain threshold (e.g. £50) is reached. Rather than complicate the paper, and bearing the word limitations in mind, we have clarified that “participants were paid equivalent to £0.60.”

37. How were the statements devised?

Response: Most of the statements referred to have now been removed from this paper, following discussions around the statistical reviewers comments. The statements asking participants why they had or had not sent their children back to school were developed based on our understanding of the literature, informal conversations with parents, our impression from meetings with policy colleagues working in this field, review of social media comments, and discussion with one school trustee.

38. PPI – do you think that one school trustee was in a position to provide pertinent feedback? Who was the trustee? Were they a parent of school-age children? Why not tap into the thoughts of a larger group of parents? I don’t understand why the questions were based on the opinion of one person alone...

Response: The trustee is one of the authors on the paper. She does have school age children. The statements were not based on the opinion of a single person – they were developed independently and then, as an additional measure, we sought a second opinion from her. In practice, in addition to our list of statements, we also included an “other – please write in” option. This was used by, at most, 13% of participants, suggesting that our statements did indeed reflect the majority of concerns that parents had.

39. How long does it take to fill in the survey? – what did you tell potential participants about this? (see supplementary material). The process looks pretty onerous to me...has this got any implications for how the form was completed?

Response: Excluding a few outliers who appear to have started to respond on one day and came back to finish the survey the next day, the median time taken to respond was 11 minutes.

40. The key information is contained in the lengthy, detailed tables and I wondered if the authors can think of an approach that might make it easier to digest this information. Would it be helpful to summarise the ‘positive’ findings in a separate table, for example?

Response: Thank you for your comment. We have moved table 1-3 into the supplementary materials and described key results narratively. We have also removed a large chunk of the more complicated analyses, which we think makes for a cleaner paper.

41. Table 1 – what about ‘I was waiting to see how other children got on first’ – is this a pertinent reason?

Response: We would hope that this is captured either under “I think it is too risky for my child to attend school at the moment” [given that the underlying motive for waiting to see is a sense of risk and the policy implications are the same] or else is something that a parent would record under the “other” option that was available to them.

42. Why did the children of keyworkers not attend school in the main?

Response: This information is in Table one of the supplementary material and in paragraph two of the Results section: “because the school was not open” (52%).

Editor(s)' Comments to Author (if any):

Associate Editor

Comments to the Author:

Thank you for the opportunity to consider this interesting and topical manuscript. Three reviewers have generously given their time to provide feedback. The consensus is that the current version of this paper requires major revision before it can be suitable for publication. Please give careful consideration to the detailed comments from each of the three reviewers and address their concerns.

43. In addition, please also provide a definition of 'willingness' in the paper and describe how this relates to behaviours and intentions.

Response: Thank you for your comment. We have now removed the data on "willingness" and refocused on actual behaviour which we think is now clearer.

We look forward to receiving the next version of your manuscript.

Editor in Chief

You need to address the comments of the stats reviewer first. This may change your findings.

You then need to address the concerns of the other two reviewers. If you need more time let us know.

44. Reviewer 2 raises concerns about too much data in the tables. We agree. Consider supplementary tables for full data and key data for tables in main paper.

Response: Thank you for your comments. We have moved table 1-3 into the supplementary materials and described key results narratively. As also suggested by reviewer 2.

45. Correct the typo in Table 4 Ethnicity BAME children of key workers % >100%

Response: Apologies, this has been corrected.

Thank you for your consideration of this manuscript.

Sincerely,

Lisa Woodland (on behalf of all authors)

VERSION 2 – REVIEW

REVIEWER	Reviewer name: Dr. Sunil Bhopal Institution and Country: Newcastle University, United Kingdom of Great Britain and Northern Ireland Competing interests: None
REVIEW RETURNED	17-Apr-2021

GENERAL COMMENTS	Congratulations to the authors on this revision and for thoroughly addressing my comments. I feel this is a much more readable and interpretable piece and hope that authors, the journal and readers agree! A couple of tiny comments I spotted which can be dealt with by the authors & editorial team. 1) BMG Research is mentioned twice (line 56 -58) but direct link given that this is the market research company mentioned above 2) Headings – seem to go askew in Methods – will be worth double checking editorially 3) Table 1 header – typo in 'workers'
--

REVIEWER	Reviewer name: Dr. Peter Flom Institution and Country: Peter Flom Consulting, United States Competing interests: None
REVIEW RETURNED	13-Apr-2021

GENERAL COMMENTS	I confine my remarks to statistical aspects of this paper. These were very simple, but appropriately so, and I recommend publication.
---

REVIEWER	Reviewer name: Dr. Tim Cheetham Institution and Country: Newcastle Upon Tyne Hospitals NHS Foundation Trust, United Kingdom of Great Britain and Northern Ireland Competing interests: None
REVIEW RETURNED	26-Apr-2021

GENERAL COMMENTS	Looking good - very minor comments only What this study adds: Is 'education' a reason not to send the child back to school? – do you mean 'perceived benefits of education'? What is BMG? Please expand when first used Clumsy grammar – eg All participants were asked to select their child with the most recent birthday to answer questions about – page 5 Why only report data from section 1?
--

VERSION 2 – AUTHOR RESPONSE

Associate Editor

Comments to the Author:

Thank you to the authors for their comprehensive reworking of the paper in response to the issues raised by the reviewers - this has resulted in a much improved manuscript focussing on behaviours.

Reviewer 1 is now satisfied with the simplified, appropriate statistics but there are a few remaining issues which have been identified by reviewers 2 and 3 - please could the authors give these careful consideration.

Reviewer: 1

Dr. Peter Flom, Peter Flom Consulting

Comments to the Author

I confine my remarks to statistical aspects of this paper. These were very simple, but appropriately so, and I recommend publication.

Response: Thank you for the recommendation.

Reviewer: 2

Dr. Sunil Bhopal, Newcastle University

Comments to the Author

Congratulations to the authors on this revision and for thoroughly addressing my comments. I feel this is a much more readable and interpretable piece and hope that authors, the journal and readers agree!

Response: Thank you for the positive feedback, we are pleased you approve of the alterations we made. We do also agree that this piece of work is much more readable.

A couple of tiny comments I spotted which can be dealt with by the authors & editorial team.

- 1) BMG Research is mentioned twice (line 56 -58) but direct link given that this is the market research company mentioned above

Response: We have moved BMG Research's citation to the first reference of the company.

- 2) Headings – seem to go askew in Methods – will be worth double checking editorially

Response: We have reviewed and clarified the headings in the methods.

- 3) Table 1 header – typo in 'workers'

Response: We have resolved.

Reviewer: 3

Dr. Tim Cheetham, Newcastle Upon Tyne Hospitals NHS Foundation Trust, Newcastle University Comments to the Author Looking good - very minor comments only

- 1) What this study adds: Is 'education' a reason not to send the child back to school? – do you mean 'perceived benefits of education'?

Response: Thank you for your comments. We have included "perceived benefits of".

- 2) What is BMG? Please expand when first used

Response: We have altered the sentences referring to BMG Research. "We commissioned a market research company, BMG Research to administer a cross-sectional survey..."[citation] and "participants (n=2,447) were recruited from BMG Research's panel and to achieve a sample broadly representative of the population, BMG Research monitored region, child age, child gender, parent/guardian age, and parent/guardian gender."

Clumsy grammar – eg All participants were asked to select their child with the most recent birthday to answer questions about – page 5

Response: We have clarified. "All participants answered questions referring to their child who had the most recent birthday."

Why only report data from section 1?

Response: We have only reported data from section 1 to make it easier to understand by focusing on actual behaviour. This has now added clarity to the manuscript as noted by reviewer 2 and the associate editor.